# The Pharmacy of the Future: Pharmacy Professionals’ Perceptions and Contributions Regarding New Services in Community Pharmacies

**DOI:** 10.3390/healthcare11182580

**Published:** 2023-09-18

**Authors:** Artemisa R. Dores, Miguel Peixoto, Irene P. Carvalho, Ângelo Jesus, Fernando Moreira, António Marques

**Affiliations:** 1Laboratório de Reabilitação Psicossocial, Centro de Investigação em Reabilitação (CIR), Escola Superior de Saúde, Instituto Politécnico do Porto (ESS-IPP), 4200-072 Porto, Portugal; mftmp@sc.ipp.pt (M.P.); ajmarques@ess.ipp.pt (A.M.); 2Clinical Neurosciences and Mental Health Department, Faculty of Medicine, University of Porto, 4200-319 Porto, Portugal; irenec@med.up.pt; 3CINTESIS@RISE, Faculty of Medicine, University of Porto, 4200-450 Porto, Portugal; 4CISA, Escola Superior de Saúde, Instituto Politécnico do Porto (ESS-IPP), 4200-072 Porto, Portugal; acj@ess.ipp.pt (Â.J.); ffm@ess.ipp.pt (F.M.)

**Keywords:** community pharmacies, services, healthcare, herb/dietary supplement–drug interactions

## Abstract

The use of information and communication technologies (ICTs) has revolutionized the provision of health services, often referred to as eHealth, benefiting community pharmacies that can offer new services in innovative formats, namely through telepharmacy. This study aimed to explore the perceptions of pharmacy professionals (i.e., pharmacists and pharmacy technicians) on the provision of new services. The study consisted of administering an online questionnaire to pharmacy professionals nationwide. The questionnaire was developed by the research team, based on focus group methodology, from which an inductive analysis led to the categories that made up the dimensions of the survey. Participants were 95 pharmacy professionals with a mean age of 33.69 years old (*SD* = 10.75). Almost 79% were women. The results show overall receptivity to the development of new services in community pharmacies. Suggestions for the development of the new services, conditions necessary for their implementation, potential obstacles, and strategies to promote adherence to the new services, among others, are identified. The knowledge thus acquired will help community pharmacies to develop innovative solutions in counselling, pharmacotherapy monitoring, and pharmacovigilance, for example, of herb/dietary supplement–drug adverse reactions and interactions. Based on this information, new services can become more accessible, namely through the use of ICTs.

## 1. Introduction

Pharmacies are the final step in the supply chain before drugs reach the consumer, and pharmacists are the most accessible healthcare professionals, providing follow-up on clients’ care [1,2]. The growth in the services provided by pharmacies accompanies societal demands for extended client care. As a result, pharmacies have been shown to impact healthcare costs positively [3], contributing to the control of several diseases, such as diabetes and hypertension [4,5,6,7,8], cardiovascular diseases [7,8], asthma [4,7], drug addiction [9], pulmonary arterial hypertension [10], chronic obstructive pulmonary disease (COPD) [4], human immunodeficiency virus (HIV) treatment [11], and chronic disease management [12]. Pharmacists have also been contributing to reaching the most fragile and vulnerable populations, namely, people living in isolation [13], elderly people [12], and people with low health literacy levels, who tend to have lower rates of adherence to treatments [14]. The COVID-19 pandemic came to underline how important the new services are within the healthcare system. When the main healthcare facilities were overwhelmed during that period, pharmacists oversaw the management of minor ailments, thus alleviating some of the load of the main healthcare facilities [15].

Other examples of possible new services emerge due to the growth in polypharmacy (i.e., the simultaneous use of different medications) combined with the use of a variety of complementary and alternative medicines (CAM) in the modern era, such as herbs or foods, which can result in potentially negative interactions between the consumed substances. Despite the potentially negative effects of different interactions, there is still a lack of studies about these interactions [16,17]. A frequent public misconception is that these CAM are completely safe and, because of that, they can be used in tandem with prescription drugs [18,19,20]. Additionally, their marketing is further enhanced by the Internet [20]. However, these products can be associated with an extensive list of adverse effects that include increased risks for various heart conditions, changes in growth hormones, and the development or worsening of psychological conditions like depression and aggression [21]. Additionally, these supplements have potentially harmful interactions with other drugs, and considering the exponential growth of this market, there are data showing a significant number of cases where these harmful interactions are occurring [21]. Discussing such aspects with a healthcare professional who can indicate the role that these supplements might play in an ongoing therapy is thus important, especially given the widespread consequences that their usage can have [19]. However, some studies show that there is still a lack of understanding about herb/dietary supplement–drug interactions (HDSIs) among pharmacists [22], whereas others show that pharmacists have a good knowledge of the effects of herbal products, including side effects and interactions, reflecting the heterogeneity of results in this field [23,24].

The aforementioned body of evidence and the necessity for pharmacies to provide a wider variety of services to better assist the population and to relieve the burden of main healthcare centers justify the need for new services in pharmacies and for further improved training in diverse areas as well as the technical solutions that are available so that better services and counselling can be delivered [22,25].

Despite the recorded benefits of new services such as these, or others, in pharmacies, there are some variables that can make the implementation and maintenance of novel services difficult. Some of these variables pertain to pharmacists’ characteristics, such as perseverance in the face of barriers, proactivity, and the training necessary for a given service to be provided [26]. The structure that supports pharmacies is also important and includes sharing information, relationships with other healthcare professionals, using new techniques, and organizational support that encompasses everything from staff training to the development of service systems to support the practice [26]. This support structure is especially important in providing pharmacists with accurate information on each client’s conditions and medical history, because not having this information makes it more difficult to provide the necessary services [27]. The availability of resources, whether physical or human, is necessary for the quality of the services provided [28]. Attention to training is especially important because training is often reported as lacking [29]. The implementation of new services also may represent an increased workload, which adds another layer of difficulty to their implementation [30]. Associated with the increased workload, lack of time can cause the disruption of the regular work schedule [7,29,31,32].

Given the wide variety of tasks that can be successfully handled by pharmacies and the lack of a clearly defined role for community pharmacists, there is a further need to develop new ways of accurately testing how to implement new and advanced services. The use of information and communication technologies (ICTs) has revolutionized the provision of health services, often referred to as eHealth, and could be used as part of the solution, allowing community pharmacies to offer new services in innovative formats, namely through telepharmacy. The aim of the present study is to assess pharmacists’ perceptions of new and advanced services and how to implement them.

## 2. Materials and Methods

### 2.1. Participants

This study comprised 95 participants recruited from 2022 to 2023 from across Portugal. All professionals with a degree in Pharmacy or Pharmaceutical Sciences or students with at least one year of experience in the community pharmacy setting, having completed their final year internship at such pharmacies, were eligible for participation. No exclusion criteria were applied.

### 2.2. Instrument

The survey consisted of a questionnaire developed specifically for this study through focus groups with pharmacy professionals representing rural and urban settings in the country and diverse services, from which an inductive analysis led to a set of categories [33]. The categories found in the focus groups led to the different dimensions that constitute the current questionnaire. After the formulation of these dimensions, the questionnaire was validated conceptually through expert discussion (construct validity) and pre-tested with a small sample under similar conditions to its final application [34]. Additionally, this process was accompanied by the method of spoken reflection or “thinking aloud” [35], which aims to analyze the content and form of the items regarding their clarity and understandability.

The questionnaire included a set of socio-demographic questions (e.g., age, gender, profession, years of service, and geographical area of work) plus 17 questions targeting perceptions about all aspects related to the implementation of new and advanced services. Sixteen questions were comprised of various items, and responses were given on a 5-point Likert scale (“totally disagree”, “disagree”, “neither agree nor disagree”, “agree”, and “totally agree”) and on a 7-point scale (“no adherence”, “very weak”, “weak”, “don’t know/no opinion”, “good”, “very good”, and “excellent”), plus on an open-ended format. The final question of the questionnaire was open-ended, where participants were asked to leave any comments/suggestions about the topic. The questionnaire was organized according to three main dimensions, namely, the roles of pharmacies, potentialities for new services, and service innovations. These dimensions were in line with the categories found in the focus group analysis [33] (cf. see Appendix A).

### 2.3. Procedure

The survey was publicized (e.g., inserted in institutional sites such as office forums and university’s and laboratory’s webpages). In addition, an invitation for the study was sent via e-mail through the National Association of Pharmacies (ANF) and the research team’s pool of contacts after reception of information about the study’s goals and after signing an online informed consent form. The data were collected through an online survey. Participation was voluntary, and data anonymity and confidentiality were ensured. The study was approved by the local research ethics committee (Ethics committee registration number: CE0093B) and complies with the Declaration of Helsinki.

### 2.4. Statistical Analysis

The statistical analysis was performed in the Statistical Package for the Social Sciences (SPSS) [36]. Descriptive analyses were performed on socio-demographic variables and on quantitative responses.

## 3. Results

Of the 95 participants with work experience in the context of community pharmacies, most were pharmacists (67.4%), and fewer were categorized as pharmacy technicians (32.7%). Together, they are called “pharmacy professionals” throughout the paper. The majority were women (75.8%). The mean age was 33.69 (*SD* = 10.75) years old, ranging from 20 to 63 years old, and the mean time of professional experience was 9.88 (*SD* = 9.05) years (Table 1).

The first set of items in the questionnaire addressed pharmacy professionals’ perceptions of the role of pharmacies, with the statement, “Overall, the role of community pharmacies involves:” (Table 2). The “totally agree/agree” response was selected by 100% for the item “Educate/Inform”, by 98.9% for “Evaluate the client’s complaint(s)”, by 97.9% for “Evaluate therapeutic adhesion”, by 96.8% for “Evaluate/Measure physiological parameters”, by 95.8% for “Monitor non-pharmacological therapy”, by 94.7% for “Monitor pharmacological therapy”, by 93.7% for “In drug-supplement interactions”, by 89.5% for “Refer to a specialized doctor”, and by 71.6% for “Register the information”. For the item on “Prescribe”, “totally disagree/disagree” was selected by 42.1%, followed by “totally agree/agree”, which was selected by 32.6%.

Regarding Table 3 for the items pertaining to community pharmacies’ advantages for new service implementation, with the statement, “Community pharmacies have the potential to include the provision of new services:” (Table 3), both “The geographical proximity that pharmacies have with the community” and “The proximity, in terms of the relationship between pharmacy professional and client” received the “totally agree/agree” response by 100% of participants. For the remaining items, the “totally agree/agree” response was selected by 88.4% for “The existence of physical spaces”, by 84.2% for “The availability of professionals” and “The current knowledge of pharmacy professionals on drug-supplement interactions”, and by 82.1% for “The existence of human resources”.

For the items on service innovation, addressed with the statement, “The advantages of including these new services for you, as a pharmacy professional, would be:” (Table 4), the “totally agree/agree” response was selected by 97.9% for “Better follow-up on the health state of the client” and for “The possibility of working in a multidisciplinary team”, by 93.8% for “Improved adequacy of the services for the client”, and by 87.5% for “Further differentiation of pharmacy professionals” and for “Better payment for new services provided”.

For the items addressing “The advantages of including these new services for pharmacies would be:” (Table 5), the “totally agree/agree” response was selected by 97.9% for “Increased perception of the utility of pharmacies by clients, and the availability of more services”, by 91.7% for “Social function”, by 87.5% for “Resource optimization”, by 81.3% for “Prestige”, and by 70.8% for “Increased profits” and “Increased competitiveness of pharmacies”.

For the items addressing “The advantages of including these new services for clients would be:” (Table 6), the “totally agree/agree” response was selected by 95.8% for “Close relationship between client and pharmacy professional”, by 94.7% for “Preventing the aggravation of the clinical condition”, by 92.6% for “Faster intervention”, by 91.6% for “Reducing the number of trips to healthcare services” and “More confidence in the services provided”, and by 87.4% for “Optimizing client time in accessing the pharmacy’s service”.

For the items addressing “The role of the pharmacy in drug-supplement interactions would imply assessing:” (Table 7), the “totally agree/agree” response was selected by 100% for “Supplement consumption” and “The risk of drug-supplement interactions that the client plans to acquire”, by 94.7% for “Previously prescribed drugs” and “Blood pressure”, by 91.6% for “Weight/height”, by 89.5% for “Cholesterol”, by 87.4% for “Triglycerides”, and by 85.3% for “Glycaemia”.

For the items addressing the statement, “The role of pharmacies in non-pharmacological therapeutics involves/should involve monitoring:” (Table 8), the “totally agree/agree” response was selected by 97.9% for “Smoking cessation”, by 96.8% for “Food diet”, by 95.8% for “Lifestyle”, and by 94.7% for “Physical activity” and “Sleep hygiene”.

For the items addressing “The conditions for implementation of new services in pharmacies are/should be:” (Table 9), the “totally agree/agree” response was selected by 98.9% for “Pharmacy professionals’ training, for example, more knowledge about medical conditions/pathologies” and “Specialization of pharmacy professionals in new conditions to be considered, particularly in drug-supplement interactions”; by 91.6% for “Clarifying legal requisites, namely about the data that can be shared”; by 87.4% for “Adequate physical space/pharmacy facilities”; and by 50.5% for “Recruitment of new specialized professionals for pharmacies”.

For the items addressing the statement, “For drug-supplement interactions:” (Table 10), the “totally agree/agree” response was selected by 96.8% for “Alerts for herb interactions, providing scientific knowledge for various healthcare professionals involved in the monitoring process, such as doctors, nutritionists, and other pharmacy professionals” and “Platform for sharing valid scientific information to support interaction alerts”; by 94.7% for “Data bank with drug-supplement interactions”; by 91.6% for “Adequate equipment and computer systems at pharmacies, for drug-supplement interaction” and “Interface with access to the data collected by the pharmacy professional”; by 80% for “Interface for other healthcare professionals who do not work in a pharmacy”; and by 65.3% for “Printer for printing a list of individual interactions to give to clients”.

For the items addressing “The difficulties in implementing new services in pharmacies are/could be:” (Table 11), the “totally agree/agree” response was selected by 94.7% for “More workload for pharmacy professionals.”; by 90.5% for “More time investment by pharmacy professionals”; by 88.4% for “More financial investment by the pharmacy”; by 83.2% for “No knowledge about the acquisition of drugs or supplements acquired in other spaces, or at other pharmacies”; by 72.6% for “Lack of training about the use of new equipment”; by 70.5% for “Lack of training about how the new services work”; by 67.4% for “The sensitivity of the data managed, namely, sharing of information between pharmacy professionals and healthcare professionals”; by 64.2% for “The sensitivity of the data managed, namely, sharing of information between pharmacies”; by 63.2% for “Lack of acceptance by clients about possible drug-supplement interactions that they take or plan to take” and “Absence of a regulatory entity for supplements”; and by 54.7% for “Lack of training for pharmacy professionals to act in drug-supplement interactions”.

For the items addressing the factors that promote adherence to new pharmacy services, with the statement, “Can promote client adherence to the new services, like:” (Table 12), the “totally agree/agree” response was selected by 95.8% for “Assuring the dissemination of the added value of the new services”, by 93.7% for “Assuring the co-payment for the new services”, by 87.4% for “Assuring the confidentiality in sharing clinical information”, and by 46.3% for “The new services being free of charges”. For the item “Providing discounts for the new services”, the most frequently selected answer was “totally disagree/disagree”, 40%, followed by 37.9% for “totally agree/agree” and by 22.1% for “neither agree nor disagree”.

## 4. Discussion

The aim of this study was to survey the perceptions of pharmacy professionals regarding several aspects related to service expansion in community pharmacies.

For the first set of questions related to the role of community pharmacies, the majority of participants agreed with most of the roles presented as options in the questionnaire, such as (in descending order) educate, evaluate the clients’ complaints, evaluate therapeutic adherence, evaluate/measure physiological parameters, monitor pharmacological and non-pharmacological therapies, refer to specialized doctors, register the clients’ information, and prescribe—the last two items had comparatively smaller percentages of total agreement. In addition to expanding the role of pharmacy professionals to various tasks, these results are in line with the tasks presented in the literature; several studies associate pharmacy professionals’ role with activities such as monitoring the therapy, promoting treatment adherence [37,38], and referring the client to an appropriate healthcare professional [39]. In this study, all participants agreed that educating/prescribing was part of what they did. In the literature, this is one of the major contributions that pharmacy professionals provide [31,39,40], with positive effects on treatment adherence [41,42]. The item that was most divisive of the sample in this study was prescribing as part of the pharmacy professionals’ role. Despite almost half of the participants disagreeing, there was still a substantial part who agreed (32.6%). This result reflects the lack of consensus in the literature about this topic. Some studies report that pharmacy professionals agree that prescribing should be their role [43] and that independently prescribing would be beneficial [44,45], whereas others emphasize pharmacy professionals’ role as re-prescribing [46]. This divide in the willingness to prescribe is based on aspects such as the knowledge that the pharmacy professional has on the client, the experience in prescribing, and the adaptability of the community pharmacy toward prescribing [47]. The product to be prescribed and the condition for which it is prescribed also play a role. For example, patients and the general population only supported pharmacy professionals’ prescriptions for specific situations (e.g., in minor ailments) [48]. Given the current lack of consensus on this topic, there is a need to further understand how pharmacy professionals perceive prescribing in relation to their role, as well as the perception of the general public and of other healthcare professionals.

For the second set of items pertaining to a specific example of a possible new service, HDSIs, participants positively agreed with the capability of pharmacies to assist in these situations. Despite the need for further training [22,49], the necessity to discuss the use of these products with a healthcare professional [19], associated with the proximity that pharmacy professionals have with the public [1,2], makes them very effective at helping in these cases.

For the set of items about the factors that give pharmacy professionals the potential to include new services, all participants agreed that the proximity, both physical and relational, that pharmacies have with the community facilitates the provision of these new services. Indeed, in previous works, this proximity has been stated as facilitating pharmacy professionals in aiding clients, those being the healthcare professionals that the public contacts most often [1,2]. Additionally, the existence of physical space and the availability of professionals/human resources were two factors for which pharmacy professionals in the current study agreed are necessary for pharmacies to have the potential to provide new services. Consideration of these two factors is important because they are often reported as lacking in the literature, which causes a barrier to service implementation [4,28,50].

In terms of possible advantages that these new services bring to pharmacy professionals, participants agreed with the various advantages in the survey. Previous works indicate that new pharmacy services allow better delivery of services, for example, through the use of telepharmacy to monitor clients during the COVID-19 pandemic and in promoting treatment adherence [12,51]. In this study, the advantages of expanding pharmacies to new services were (in descending order of percentage of agreement) the availability of more services and increased utility of pharmacies as perceived by the clients, pharmacies’ social function, resource optimization, prestige, and, finally, increased profits and competitiveness. Pharmacy professionals’ perceptions of the advantages that new services bring to the pharmacies are thus in line with reports suggesting that new services help a pharmacy to be more competitive [26]. In the example of helping HDSIs cases, pharmacy professionals agreed with the various tasks necessary to monitor a variety of variables, namely, the client’s supplement/herb consumption and the risk of HDSIs, previously acquired drugs, weight/height, cholesterol, triglycerides, and glycaemia, thus highlighting the need for more training in this area [22,49]. Concerning non-pharmacological services, pharmacy professionals agreed that the pharmacy should monitor lifestyle, physical activity, food diet, smoking cessation, and sleep hygiene. These services are also associated with pharmacies in previous research [52,53], especially smoking cessation [54].

Regarding the conditions for implementing new services, most of the participants agreed with the conditions presented in the questionnaire. Two of the three chief conditions agreed upon were related to pharmacy professionals’ training and specialization. Overall, specialized training is a condition recognized as necessary for implementing new services [30,55]. The need for training seems to be pressing [22,49], and recognizing this is an important step toward the implementation of a new service. The third chief condition for the implementation of new services pertained to legal requisites, which can constrain how the pharmacy operates, as well as the established guidelines, as was recognized in previous reports [26,29]. Adequate physical space was a condition reported as necessary for new services and was also found in the literature [4]. Another condition for half of the sample was the recruitment of specialized personnel.

Participants agreed that the necessary requirements for helping (e.g., in HDSI cases) were related to technological computer systems that allow the access and sharing of information across various healthcare professionals. Such systems make it easier to access information and result in more continued and accurate monitoring of the clients. Participants agreed that having a platform that allows the dissemination of scientifically validated information on health issues would enable them to better assist their clients. These remote platforms have been increasingly present in pharmacies [56] and have shown various advantages during the COVID-19 pandemic, namely, with the use of telepharmacy [12]. However, participants additionally reported the importance of other aspects, such as the presence of alerts (e.g., for drug interactions) for various healthcare professionals involved in the monitoring process, and the existence of a data bank (e.g., with drug-supplement interactions).

For the items related to difficulties in the implementation of new pharmacy services, participants overall agreed with the difficulties presented in the survey. In descending order, an increased workload (with the implementation of new services) was the most frequently selected difficulty, which is corroborated by the literature [30]. For example, with new services, additional time investment might be required due to the potential disruption of the normal work schedule. Financial investment by the pharmacies was a difficulty also agreed upon by the vast majority of the participants, and, indeed, financial constraints are shown to make the implementation of a client-centered approach difficult [2]. To a lesser extent, the majority of participants also agreed that lack of training on how the new services work and on the use of new equipment constitute a difficulty in the implementation of new services. More than half of the sample also agreed that sharing sensitive data between pharmacy professionals, other healthcare professionals, and other pharmacies was difficult, namely because there are various regulations that need to be observed so that client data can be properly managed and stored [57].

Concerning factors that promote clients’ adherence to new pharmacy services, dissemination of the added value of the new services was selected by almost the entire sample (95.8%). This plays into clients’ perceptions because if the new services are seen as being effective for the client’s condition, then greater adherence follows [58]. Additionally, the presence of expanded services increases the client’s expectancy about the organization’s professional quality and leads to a more positive reaction to said services [59]. Co-payment of the services was also selected by the vast majority of participants as a factor that would help adherence, whereas many participants (40%) disagreed that providing discounts (which is another form of financial aid) increases adherence. There is still no significant literature on the topic of discounts and service adherence. One study [60] showed that if both physician and client receive financial incentives, treatment adherence is enhanced. Perhaps the relationship between providing a discount and adherence is not linear, and more investigation is necessary to determine whether discounts have an actual effect. Regarding participants’ answers as to whether or not the new services would be free of charge to promote adherence, the results were also divided, with almost half of the participants agreeing that it would help, whereas many disagreed. Nevertheless, new services that have a discount or are free of charge could aid with the problem that some clients have about not being able to pay for a continued service [27], which, in turn, could contribute toward service adherence. For the final item (ensuring confidentiality), almost 90% of participants agreed that this was an adherence-promoting factor. Some previous studies also reported that lack of privacy has a negative influence on adherence to services [61,62]. In their study [27], the authors showed a practical example of the importance of confidentiality by way of a client who reported that the pharmacies shared the treatment with the client’s family.

In sum, pharmacies are increasingly fundamental structures of the healthcare system, as they hold the possibility of offering new health services due to their wide diffusion and proximity to the community. In addition to dispensing medicinal products, qualified health professionals, such as pharmacy professionals, can advise patients on drug regimens, herbal and dietary supplements, and HDSIs and provide pharmacovigilance services, among others. The close contact that exists between pharmacies and the population presents a great opportunity for the public healthcare system in view of the fact that pharmacies are becoming first-layer healthcare actors.

This study contributed to a better understanding of how pharmacy professionals perceive their role and how they perceive their capabilities to address health aspects that, in the literature, are associated with community pharmacies’ positive impact. Participants recognize the advantages of new services and of pharmacies’ potential to host them, but the development of new services is not without difficulties. There is agreement among participants about the several factors that play into establishing a new pharmacy service, both as regards conditions and difficulties. Some of these factors are related directly to pharmacies and pharmacists’ roles, whereas others are associated with broader aspects, such as the law and lawmakers. These issues gathered significant percentages of agreement on the part of pharmacy professionals in this study. However, the role that other healthcare professionals and the clients have in the success of new pharmacy services should not be forgotten, and knowledge of their perceptions about these various aspects is also necessary. More dissemination of pharmacy professionals’ capabilities is also needed for the public perspective to change and possibly also for pharmacy professionals’ acceptability of new roles to increase. The changes in pharmacy professionals’ roles should also come with increased training to capacitate them to deliver the various services.

Despite the results found in this study, there are still various questions that need to be addressed so that the process of successfully carrying out a service expansion in pharmacies is fully understood. This study only covered one of the parties involved in service expansion, which is insufficient to accurately determine the points that should be addressed for service innovation. The perspectives of other healthcare professionals and clients should also be explored, as mentioned earlier. The goal of this study was to provide the view of pharmacy professionals in general about service expansion. It would be interesting, in future research, to also inspect potential group differences in perspectives (e.g., between pharmacists and pharmacy technicians). The use of information and communication technologies (ICTs) and artificial intelligence (IA) are among the new services that can be implemented in pharmacies.

## 5. Conclusions

This study showed pharmacy professionals’ receptivity to the development of new services in community pharmacies. The results allowed the identification of suggestions for the development of new pharmacy services, conditions necessary for their implementation, potential obstacles, and strategies to promote adherence to new services. These results will help community pharmacies to develop innovative solutions in counselling, pharmacotherapy monitoring, and pharmacovigilance.

## Figures and Tables

**Table 1 healthcare-11-02580-t001:** Socio-demographic characteristics.

Socio-Demographic Characteristics	*n* (%)	*M* (*SD*)	Range Min–Max
Female	72 (75.8)	-	-
Male	23 (24.2)	-	-
Pharmacist	64 (67.4)	-	-
Pharmacy Technician	31 (32.7)	-	-
Age	-	33.69 (10.75)	20–63
Years of experience	-	9.93 (8.91)	1–35

**Table 2 healthcare-11-02580-t002:** Community pharmacy involvement in various tasks (*N* = 95).

Items	*M* (*SD*)	Range Min–Max	Totally Disagree/Disagree *n* (%)	Neither Agree nor Disagree *n* (%)	Totally Agree/Agree *n* (%)
Overall, the role of community pharmacies involves:					
Evaluate the client’s complaint(s)	2.99 (0.10)	2–3	0 (0.0)	1 (1.1)	94 (98.9)
Evaluate therapeutic adhesion	2.98 (0.14)	2–3	0 (0.0)	2 (2.1)	93 (97.9)
Evaluate/Measure physiological parameters	2.97 (0.76)	2–3	0 (0.0)	3 (3.2)	92 (96.8)
Refer to a specialized doctor	2.85 (0.46)	1–3	4 (4.2)	6 (6.3)	85 (89.5)
Monitor pharmacological therapy	2.94 (0.29)	1–3	1 (1.1)	4 (4.2)	90 (94.7)
Monitor non-pharmacological therapy	2.94 (0.28)	1–3	1 (1.1)	3 (3.2)	91 (95.8)
Educate/Inform	3.00 (0.00)	3–3	0 (0.0)	0 (0.0)	95 (100)
Prescribe	1.91 (0.86)	1–3	40 (42.1)	24 (25.3)	31 (32.6)
Register the information	2.67 (0.55)	1–3	4 (4.2)	23 (24.2)	68 (71.6)
Provision of services for drug-supplement interactions	2.93 (0.30)	1–3	1 (1.1)	5 (5.3)	89 (93.7)

**Table 3 healthcare-11-02580-t003:** Community pharmacies’ advantages for new service implementation (*N* = 95).

Items	*M* (*SD*)	Range Min–Max	Totally Disagree/Disagree *n* (%)	Neither Agree nor Disagree *n* (%)	Totally Agree/Agree *n* (%)
Community pharmacies have the potential to include new services, given:					
The geographical proximity that pharmacies have with the community	3.00 (0.00)	3–3	0 (0.0)	0 (0.0)	95 (100)
The proximity, in terms of relationship between pharmacy professional and client	3.00 (0.00)	3–3	0 (0.0)	0 (0.0)	95 (100)
The existence of physical spaces	2.86 (0.40)	1–3	2 (2.1)	9 (9.5)	84 (88.4)
The existence of human resources	2.75 (0.58)	1–3	7 (7.4)	10 (10.5)	78 (82.1)
The availability of professionals	2.80 (0.50)	1–3	4 (4.2)	11 (11.6)	80 (84.2)
The current knowledge of pharmacy professionals on drug-supplement interaction	2.82 (0.44)	1–3	2 (2.1)	13 (13.7)	80 (84.2)

**Table 4 healthcare-11-02580-t004:** Advantages of the new services for pharmacy professionals (*N* = 48).

Items	*M* (*SD*)	Range Min–Max	Totally Disagree/Disagree *n* (%)	Neither Agree nor Disagree *n* (%)	Totally Agree/Agree *n* (%)
The advantages of including these new services for you, as a pharmacy professional would be:					
Improved adequacy of the services for the client	2.94 (0.25)	2–3	0 (0.0)	3 (6.3)	45 (93.8)
Better follow-up on the health state of the client	2.98 (0.14)	2–3	0 (0.0)	1 (2.1)	47 (97.9)
Further differentiation of pharmacy professionals	2.88 (0.33)	2–3	0 (0.0)	6 (12.5)	42 (87.5)
Better payment for new services provided	2.88 (0.33)	2–3	0 (0.0)	6 (12.5)	42 (87.5)
The possibility of working in a multidisciplinary team	2.98 (0.14)	2–3	0 (0.0)	1 (2.1)	47 (97.9)

Note. The smaller number of participants in this question is due to some pharmacy professionals mistakenly responding to a version of the survey that lacked this particular question.

**Table 5 healthcare-11-02580-t005:** Advantages of the new services for community pharmacies (*N* = 48).

Items	*M* (*SD*)	Range Min–Max	Totally Disagree/Disagree *n* (%)	Neither Agree nor Disagree *n* (%)	Totally Agree/Agree *n* (%)
The advantages of including these new services for pharmacies would be:					
Increased perception of the utility of pharmacies by clients, and the availability of more services	2.98 (0.14)	2–3	0 (0.0)	1 (2.1)	47 (97.9)
Increased profits	2.69 (0.51)	1–3	1 (2.1)	13 (27.1)	34 (70.8)
Prestige	2.77 (0.52)	1–3	2 (4.2)	7 (14.6)	39 (81.3)
Social function	2.92 (0.28)	2–3	0 (0.0)	4 (8.3)	44 (91.7)
Resource optimization	2.85 (0.41)	1–3	1 (2.1)	5 (10.4)	42 (87.5)
Increased competitiveness of pharmacies	2.65 (0.60)	1–3	3 (6.3)	11 (22.9)	34 (70.8)

Note. The smaller number of participants in this question is due to some pharmacy professionals mistakenly responding to a version of the survey that lacked this particular question.

**Table 6 healthcare-11-02580-t006:** Advantages of the new services for clients (*N* = 95).

Items	*M* (*SD*)	Range Min–Max	Totally Disagree/Disagree *n* (%)	Neither Agree nor Disagree *n* (%)	Totally Agree/Agree *n* (%)
The advantages of including these new services for clients would be:					
Reducing the number of trips to healthcare services	2.87 (0.44)	1–3	4 (4.2)	4 (4.2)	87 (91.6)
Preventing the aggravation of the clinical condition	2.95 (0.23)	2–3	0 (0.0)	5 (5.3)	90 (94.7)
Close relationship between client and pharmacy professional	2.94 (0.32)	1–3	2 (2.1)	2 (2.1)	91 (95.8)
More confidence in the services provided	2.91 (0.33)	1–3	1 (1.1)	7 (7.4)	87 (91.6)
Faster intervention	2.91 (0.36)	1–3	2 (2.1)	5 (5.3)	88 (92.6)
Optimizing client time in accessing the pharmacy’s service	2.83 (0.48)	1–3	4 (4.2)	8 (8.4)	83 (87.4)

**Table 7 healthcare-11-02580-t007:** Variables related to drug–supplement interactions that could be assessed by pharmacies (*N* = 95).

Items	*M* (*SD*)	Range Min–Max	Totally Disagree/Disagree *n* (%)	Neither Agree nor Disagree *n* (%)	Totally Agree/Agree *n* (%)
The role of the pharmacy in drug-supplement interaction would imply assessing:					
Previously prescribed drugs	2.94 (0.29)	1–3	1 (1.1)	4 (4.2)	90 (94.7)
Supplement consumption	3 (0.00)	3–3	0 (0.0)	0 (0.0)	95 (100)
The risk of drug-supplement interactions that the client plans to acquire	3 (0.00)	3–3	0 (0.0)	0 (0.0)	95 (100)
Glycaemia	2.81 (0.49)	1–3	4 (4.2)	10 (10.5)	81 (85.3)
Cholesterol	2.86 (0.43)	1–3	3 (3.2)	7 (7.4)	85 (89.5)
Triglycerides	2.84 (0.45)	1–3	3 (3.2)	9 (9.5)	83 (87.4)
Weight/height	2.88 (0.41)	1–3	3 (3.2)	5 (5.3)	87 (91.6)
Blood pressure	2.93 (0.33)	1–3	2 (2.1)	3 (3.2)	90 (94.7)

**Table 8 healthcare-11-02580-t008:** Non-pharmacological therapeutic monitoring (*N* = 95).

Items	*M* (*SD*)	Range Min–Max	Totally Disagree/Disagree *n* (%)	Neither Agree nor Disagree *n* (%)	Totally Agree/Agree *n* (%)
The role of pharmacies in non-pharmacological therapeutics involves/should involve monitoring:					
Physical activity	2.95 (0.23)	2–3	0 (0.0)	5 (5.3)	90 (94.7)
Lifestyle	2.96 (0.20)	2–3	0 (0.0)	4 (4.2)	91 (95.8)
Food diet	2.97 (0.18)	2–3	0 (0.0)	3 (3.2)	92 (96.8)
Smoking cessation	2.98 (0.14)	2–3	0 (0.0)	2 (2.1)	93 (97.9)
Sleep hygiene	2.95 (0.22)	2–3	0 (0.0)	5 (5.3)	90 (94.7)

**Table 9 healthcare-11-02580-t009:** Conditions for service implementation (*N* = 95).

Items	*M* (*SD*)	Range Min–Max	Totally Disagree/Disagree *n* (%)	Neither Agree nor Disagree *n* (%)	Totally Agree/Agree *n* (%)
The conditions for implementation of new services in pharmacies are/should be:					
Clarifying legal requisites, namely about the data that can be shared	2.90 (0.37)	1–3	2 (2.1)	6 (6.3)	87 (91.6)
Pharmacy professionals’ training, for example, more knowledge about medical conditions/pathologies	3.00 (0.21)	1–3	1 (1.1)	0 (0.0)	94 (98.9)
Specialization of pharmacy professionals in new conditions to be considered, particularly in drug-supplement interaction	2.98 (0.21)	1–3	1 (1.1)	0 (0.0)	94 (98.9)
Recruitment of new specialized professionals for pharmacies	2.37 (0.72)	1–3	13 (13.7)	34 (35.8)	48 (50.5)
Adequate physical space/pharmacy facilities	2.84 (0.45)	1–3	3 (3.2)	9 (9.5)	83 (87.4)

**Table 10 healthcare-11-02580-t010:** Conditions for helping with drug–supplements interactions (*N* = 95).

Items	*M* (*SD*)	Range Min–Max	Totally Disagree/Disagree *n* (%)	Neither Agree nor Disagree *n* (%)	Totally Agree/Agree *n* (%)
For drug-supplement interactions:					
Adequate equipment and computer systems at pharmacies for drug-supplement interaction	2.91 (0.33)	1–3	1 (1.1)	7 (7.4)	87 (91.6)
Printer for printing a list of individual interactions to give to clients	2.55 (0.68)	1–3	10 (10.5)	23 (24.2)	62 (65.3)
Interface with access to the data collected by the pharmacy professional	2.92 (0.28)	2–3	0 (0.0)	8 (8.4)	87 (91.6)
Data bank with drug-supplement interactions	2.95 (0.23)	2–3	0 (0.0)	5 (5.3)	90 (94.7)
Interface for other healthcare professionals that do not work in a pharmacy	2.78 (0.47)	1–3	2 (2.1)	17 (17.9)	76 (80)
Alerts for herb interactions, providing scientific knowledge for various healthcare professionals involved in the monitoring, such as doctors, nutritionists, and other pharmacy professionals	2.97 (0.18)	2–3	0 (0.0)	3 (3.2)	92 (96.8)
Platform for sharing valid scientific information to support interaction alerts	2.97 (0.18)	2–3	0 (0.0)	3 (3.2)	92 (96.8)

**Table 11 healthcare-11-02580-t011:** Difficulties in new service implementation (*N* = 95).

Items	*M* (*SD*)	Range Min–Max	Totally Disagree/Disagree *n* (%)	Neither Agree nor Disagree *n* (%)	Totally Agree/Agree *n* (%)
The difficulties in implementing new services in pharmacies are/could be:					
More workload for pharmacy professionals	2.93 (0.33)	1–3	2 (2.1)	3 (3.2)	90 (94.7)
More financial investment by the pharmacy	2.86 (0.40)	1–3	2 (2.1)	9 (9.5)	84 (88.4)
More time investment by pharmacy professionals	2.87 (0.42)	1–3	3 (3.2)	6 (6.3)	86 (90.5)
The sensitivity of the data managed, namely the sharing of information between pharmacy professionals and healthcare professionals	2.50 (0.78)	1–3	17 (17.9)	14 (14.7)	64 (67.4)
The sensitivity of the data managed, namely the sharing of information between pharmacies	2.46 (0.78)	1–3	17 (17.9)	17 (17.9)	61 (64.2)
Lack of training about the use of new equipment	2.59 (0.72)	1–3	13 (13.7)	13 (13.7)	69 (72.6)
Lack of training about how the new services work	2.56 (0.74)	1–3	14 (14.7)	14 (14.7)	67 (70.5)
Lack of training for pharmacy professionals to act in drug-supplement interactions	2.28 (0.86)	1–3	25 (26.3)	18 (18.9)	52 (54.7)
Lack of acceptance by clients about possible drug-supplement interactions that they take or plans to take	2.45 (0.78)	1–3	17 (17.9)	18 (18.9)	60 (63.2)
Absence of a regulatory entity for supplements	2.50 (0.73)	1–3	13 (13.7)	22 (23.2)	60 (63.2)
No knowledge about the acquisition of drugs or supplements acquired in other spaces, or at other pharmacies	2.76 (0.58)	1–3	7 (7.4)	9 (9.5)	79 (83.2)

**Table 12 healthcare-11-02580-t012:** Client adherence-promoting factors to the new services (*N* = 95).

Items	*M* (*SD*)	Range Min–Max	Totally Disagree/Disagree *n* (%)	Neither Agree nor Disagree *n* (%)	Totally Agree/Agree *n* (%)
Can promote client adherence to the new services, like:					
Providing discounts for the new services	1.98 (0.89)	1–3	38 (40)	21 (22.1)	36 (37.9)
The new services being free of charges	2.12 (0.90)	1–3	33 (34.7)	18 (18.9)	44 (46.3)
Assuring the co-payment for the new services	2.93 (0.30)	1–3	1 (1.1)	5 (5.3)	89 (93.7)
Assuring the dissemination of the added value of the new services	2.96 (0.20)	2–3	0 (0.0)	4 (4.2)	91 (95.8)
Assuring the confidentiality in sharing clinical information	2.86 (0.38)	1–3	1 (1.1)	11 (11.6)	83 (87.4)

## Data Availability

Not applicable.

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
