# Peer review of "The Pharmacy of the Future: Pharmacy Professionals’ Perceptions and Contributions Regarding New Services in Community Pharmacies"

_healthcare, 2023, doi:10.3390/healthcare11182580_

Round 1

Reviewer 1 Report

See document!

Author Response

We appreciate the feedback about our manuscript that resulted in valuable improvements. We hope that the responses provided below and changes in the paper (with track changes) can address your concerns.

Kind Regards,

The Corresponding Author

Reviewer 1

Comments 1 (Reviewer 1): The title is not a good representation of the article. Please modify to specify the service being asked about. Suggestion: Pharmacy professionals’ receptivity to the development of herbs/dietary supplements-drug interactions (HDSIs) services in community pharmacies

 Response 1: We added a third title option (in the manuscript) that combines elements of the original with the suggested title.

Comments 2 (Reviewer 1): Thank you for the opportunity to review your article! I am very interested in pharmacy service development and implementation, so I feel like your survey asked about many of the main items that need to be thought of. I appreciate the construction and validation of the survey. Overall, I think this provides a summary of items to consider when starting a new service, but I would have liked the survey to either be general about services or keep it specific to HDSIs. Some of the question stems presented in the tables made it confusing if respondents knew what they were answering. It may be helpful to make your complete survey available as an appendix.

 Response 2: Our goal in this paper was to obtain a more general view on new services, and the HDSIs served merely as an example. In a previous work, we specifically addressed HDSIs. The current paper provides a more general view on new services while complementing previous research on HDSIs.  

The complete survey is now made available as an appendix for further clarification.

Comments 3 (Reviewer 1): The first two sentences should be reduced to one and added to the background.

Response 3: Text was revised according to the comment.

Comments 4 (Reviewer 1): Results should include number of respondents and a few highlighted results.

Response 4: The number of respondents and their characteristics are now included in the abstract, following this suggestion.

Comments 5 (Reviewer 1): The intro is very long. Reduce to 1 page to avoid duplication.

Response 5: The Introduction has been shortened to 1 page, as suggested.

Comments 6 (Reviewer 1): Table 1: identify what “TSDT” is

Response 6: The term TSDT has been eliminated and now only one category is used (pharmacy technician) for simplification.

Comments 7 (Reviewer 1): What was the survey response rate?

Response 7: We did not include this rate because the survey was made available to the public within the pharmacy areas, in addition to the e-mails sent via the National Association of Pharmacies, without a specified initial N.

Comments 8 (Reviewer 1): Procedure: how were participants identified and asked to consent?

 Response 8: Further explanation was added to the text to clarify how paticipants were recruited and asked to consent.

Comments 9 (Reviewer 1): When did this occur? What were the inclusion/exclusion criteria (age, pharmacy setting, etc.)? Please clarify process in your methods

Response 9: The process occurred during 2022-2023. The inclusion/exclusion criteria were having a degree in Pharmacy or Pharmaceutical Sciences, or having some experience in the community pharmacy setting (e.g., students in their final year, having completed their internship at pharmacies at least for a year were also eligible). This information has been included in the text.

Comments 10 (Reviewer 1): Lines 130-137 are discussing how the instrument was created and validated. I believe this would fit better under “Instrument” and move this section to the top of the methods since this happened before distribution.

Response 10: We have included the mentioned text under “instruments”. However, this subsection (instruments) appears after the “Participants”, as is customary.

Comments 11 (Reviewer 1): Please elaborate on how focus groups were selected, and number of individuals with demographics to make sure it aligns with the study population.

Response 11: The focus groups were conducted with pharmacy professionals representing rural and urban settings in the country and diverse services. This information has been included in the text. The details of the focus groups are the object of a different paper (Dores, A. R., Peixoto, M., Carvalho, I. P., Castro, M., & Marques, A., 2023, in press).

Comments 12 (Reviewer 1): Statistical Analysis: how did you determine the open-ended responses had no value? Why were p-values not provided to determine if there were differences in any answers?

Response 12: We meant to say that no new information emerged within the category “others” in the survey. To avoid this confusion, we have eliminated references to the category “others” from the text. This paper was aimed at providing a descriptive, general view, and highlight the perspectives of pharmacy professionals on these various topics. Analyzing differences between responses would not contribute toward these goals. Also, the paper has been already considered as too extensive.

Comments 13 (Reviewer 1): Stats should be run to make sure that there were no differences between pharmacists and pharmacy technicians since all results are reported with everyone grouped. If there is no difference, that should be stated to explain why they were grouped. If there is a difference, the tables and results will need to be updated. An update to the method’s section as described above would help me to understand if student, TSDT, and teacher should be included in results. Since there is such a small response from these groups, I am not sure that it is appropriate to include, but it would depend on the study methods.

Response 13: Comparisons between these groups are pertinent and could be the focus of a future paper. The goal of the current paper is to improve pharmacy services according to pharmacy professionals’ overall views, regardless of their type or category. We now have added this point in the study’s limitations. The aforementioned groups (teacher, student, and TSDT) are now all incorporated into pharmacy technician because their academic basis is in this area and they have experience in the community pharmacy setting.

Comments 14 (Reviewer 1): Were other services asked about? Some questions are general to all services, but the aim of the study was for one specific service. Consider summarizing general question results in the text and focusing on the specific service in the tables

Response 14: Our aim was to assess new services in pharmacies in general, and HDSIs were used as an example. We have removed specific reference to HDSIs in the abstract to avoid this confusion.

Comments 15 (Reviewer 1): Way too many tables- need to combine or move some to appendix

Response 15: We have proceeded accordingly, and have combined and removed some tables.

Comments 16 (Reviewer 1): Results text should supplement and/or summarize the table information, but yours is duplicate.

Response 16: The information in the text highlights mainly the extreme values in the tables, to simplify the interpretation of the data.

Comments 17 (Reviewer 1): Tables 2-17: remove question stem from the title and put into the far left column of each table. Modify title of tables to be reflective of what it is showing

Response 17: We have altered the tables accordingly.

Comments 18 (Reviewer 1): Table 5: Did only pharmacists get these questions? Or was there a large non- response to these questions? Please address appropriately in Methods/Discussion

Response 18: A few pharmacy professionals mistakenly responded to a slightly different version of the survey (that had been developed for non-pharmacy health professionals) that lacked this particular question. Because the questionnaire was otherwise the same (except for the lack of a few questions in the one for non-pharmacy health professionals), those pharmacy professionals’ responses were included in the analyses.

Comments 19 (Reviewer 1): Are any of these items statistically significant? No p-values are given

Response 19: Please, see our responses to Comments 12 and 13.

Comments 20 (Reviewer 1): Table 15: this statement does not make sense as written, “Do you consider that the use new services can/could be activated:”

Response 20: Table 15 was removed to avoid this confusion (and to shorten the paper, as suggested). The respective text was also removed.

Comments 21 (Reviewer 1): I do not understand this question, “How would you classify the adherence of the different stakeholders to these new services:”

Response 21: This table was also removed to avoid this confusion (and to shorten the paper, as suggested). The respective text was also removed.

Comments 22 (Reviewer 1): The discussion is very long. Consider summarizing points more concisely.

Response 22: We have shortened the discussion by removing the parts pertaining to the removed results.

Comments 23 (Reviewer 1): 368-370 is missing a third factor: “Another three factors about which pharmacists in the current study agreed upon for pharmacies to have the potential to provide new services were the existence of physical space, the availability of professionals/human resources.

Response 23: The text reads, “Two of the three chief conditions agreed upon were related to pharmacy professionals’ training and specialization … The third chief condition for the implementation of new services pertained to legal requisites”.

Comments 24 (Reviewer 1): 417 is missing a third factor: Three conditions that came up were related to training and specialization in HDSIs.

Response 24: This sentence was excluded, thank you again for this observation.

Comments 25 (Reviewer 1): 484-485 should services be added as highlighted: Concerning factors that promote clients’ adherence to new pharmacy services, dissemination of the added value of the new services was selected by almost the entire sample.

Response 25: We have added “services” to the sentence, thank you.

Comments 26 (Reviewer 1): Discuss why there was a very small response from teacher/student.

Response 26: These professional categories were merged into one (pharmacy technician), thus those subgroups have disappeared.

Comments 27 (Reviewer 1): I think the first paragraph of the conclusion sufficiently summarizes the main points of the paper.

Response 27: We agree, and the remaining text was eliminated accordingly, thank you for this observation.

Reviewer 2 Report

Overall

Thank you for the opportunity to read this manuscript. This manuscript provides a lot of information on what the authors were investigating.

One general concern I have with this manuscript is how broad some of the descriptions can be regarding the study. The focus of the background section seems to be on drug interactions with herbs and dietary supplements and training to improve education on those interactions. However, the questionnaire used goes far beyond what was discussed in the background. Simplifying everything to focusing on this education initiative will substantially strengthen this manuscript.

Reading through this manuscript, there is a lot of information to the point I think it unfortunately will overwhelm readers. There was so much data collected and presented, it is hard to keep straight if the focus is specifically on HDSI intervention role expansion of if that role is merely one used to generally described pharmacy role expansion.

Abstract

Abstract does a good job introducing the project, how telepharmacy has grown, and what opportunities exist for practice now.

I would create new lines for each of the sections of the abstract or get rid of section headers altogether. Having section headers I think is beneficial, but can make reading harder if they don’t start a new line of the abstract.  

Introduction

p.1, line 36. “Most accessible” might be a more appropriate description compared to “most frequently seen”.

p.1, line 43. “have also reached” instead of “have also been contributing to reaching”. Just makes the point more succinctly.

I am confused by the flow of paragraphs 2, 3, and 4. Paragraph 2 talks about “herbs and dietary supplements” and the importance of understanding. In paragraph 3, it is unclear if these are still what is being referred to when the paper shifts to talking about IPEDs (especially since the “drug” name in IPEDs would not include herbs and dietary supplements in most pharmacists’ definitions). Then in paragraph 4, the verbiage switches again to “complementary and alternative medicine”, which reads to be the same thing, but the rapid changes in nomenclature could read confusing to some. Also, I think the point made in paragraph 2 beginning on line 50, fits better in the paragraph beginning on line 74 to combine how these interactions manifest themselves.

Materials and Methods

p. 3, line 119. Which country? Since this will have international readership, it is important to be precise with which country so readers have an idea of how the practice of pharmacy might look different.

Why were pharmacy technicians included? Having additional opportunities for technicians is admirable, but technicians could be more limited than they are in the sampled population.

I would not include pharmacists and technicians under the single banner of “pharmacists”. If technicians are included, it might read better to say “Pharmacy staff” or “Pharmacy workforce” instead of just “pharmacists” like is mentioned on line 120. Since questions seem to be about pharmacies instead of any one area of the workforce, a measurement of differences between pharmacists and technicians would be interesting.

Results

When formatting, put some additional space between the ends of paragraphs before the table title and some additional space below the table. It could help make everything cleaner.  

Like I mentioned above, I don’t know if enough groundwork was laid in the background for all the items being surveyed on within this study. The manuscript could be strengthened by simplifying from all potential added services. Focus on the HDSI services in table titles so it doesn’t seem like other services were just dropped through the survey.

Simplify results to what is important. 17 tables is hard to keep everything straight and make sure that the question the reader is thinking of is actually the table they are looking to reference.

Discussion

Lines 326 and 327 talk about “several aspects related to service expansion”, but it would be better to put the focus on what based on the background and the service included in the results of the HDSI in focus here.

There’s a lot of information which was gathered in the survey, to the point that even in the discussion, I don’t believe readers will be able to keep everything straight. All the information either needs to be boiled down to what are the most important factors and simplified that way or needs to be broken up into multiple manuscripts. Is the focus on new services in general and what goes into that or is the focus on this particular new service, because it can go back and forth.

Conclusion

Conclusion touches on everything which has been covered in the previous sections of manuscript.

A few changes listed above, but generally good quality grammar.

Author Response

We appreciate the feedback about our manuscript that resulted in valuable improvements. We hope that the responses provided below and changes in the paper (with track changes) can address your concerns.

Kind Regards,

The Corresponding Author

Reviewer 2

Comment 1 (Reviewer 2): One general concern I have with this manuscript is how broad some of the descriptions can be regarding the study. The focus of the background section seems to be on drug interactions with herbs and dietary supplements and training to improve education on those interactions. However, the questionnaire used goes far beyond what was discussed in the background. Simplifying everything to focusing on this education initiative will substantially strengthen this manuscript.

Response 1: The paper does focus on broader aspects and on education initiatives, and the text has been modified to clarify that interactions with herbs and dietary supplements serve as mere examples, among other services. In an attempt to make the introduction more concise we have removed some of the references to HDSIs.

Comment 2 (Reviewer 2): Reading through this manuscript, there is a lot of information to the point I think it unfortunately will overwhelm readers. There was so much data collected and presented, it is hard to keep straight if the focus is specifically on HDSI intervention role expansion of if that role is merely one used to generally described pharmacy role expansion.

Response 2: The focus in on new services in general, and HDSIs are used to generally described pharmacy role expansion. Hopefully the previously mentioned modifications that we have conducted in the text will make this aspect clearer.

Comment 3 (Reviewer 2): Abstract does a good job introducing the project, how telepharmacy has grown, and what opportunities exist for practice now.

Response 3: Thank you for your comment.

Comment 4 (Reviewer 2): I would create new lines for each of the sections of the abstract or get rid of section headers altogether. Having section headers I think is beneficial, but can make reading harder if they don’t start a new line of the abstract.

Response 4: We deleted the headers to make reading easier.

Comment 5 (Reviewer 2): p.1, line 36. “Most accessible” might be a more appropriate description compared to “most frequently seen”.

Response 5: We have made the suggested change.

Comment 6 (Reviewer 2): p.1, line 43. “have also reached” instead of “have also been

contributing to reaching”. Just makes the point more succinctly.

Response 6: We have made the suggested change.

Comment 7 (Reviewer 2): I am confused by the flow of paragraphs 2, 3, and 4. Paragraph 2 talks about “herbs and dietary supplements” and the importance of understanding. In paragraph 3, it is unclear if these are still what is being referred to when the paper shifts to talking about IPEDs (especially since the “drug” name in IPEDs would not include herbs and dietary supplements in most pharmacists’ definitions). Then in paragraph 4, the verbiage switches again to “complementary and alternative medicine”, which reads to be the same thing, but the rapid changes in nomenclature could read confusing to some. Also, I think the point made in paragraph 2 beginning on line 50, fits better in the paragraph beginning on line 74 to combine how these interactions manifest themselves.

Response 7: We have adjusted the introduction to be more concise and in a more understandable order, and eliminated some parts, namely those starting on line 50 of paragraph 2.

Comment 8 (Reviewer 2): p. 3, line 119. Which country? Since this will have international readership, it is important to be precise with which country so readers have an idea of how the practice of pharmacy might look different.

Response 8: We have included the country (Portugal) in the text, as suggested.

Comment 9 (Reviewer 2): Why were pharmacy technicians included? Having additional opportunities for technicians is admirable, but technicians could be more limited than they are in the sampled population.

Response 9: The goal of this paper was to obtain the perspectives of pharmacy professionals with actual work experience in community pharmacies (namely, interacting with the public), regardless of their specific area of knowledge.

Comment 10 (Reviewer 2): I would not include pharmacists and technicians under the single banner of “pharmacists”. If technicians are included, it might read better to say “Pharmacy staff” or “Pharmacy workforce” instead of just “pharmacists” like is mentioned on line 120. Since questions seem to be about pharmacies instead of any one area of the workforce, a measurement of differences between pharmacists and technicians would be interesting.

Response 10: We have renamed the group as “pharmacy professionals” to include pharmacist and pharmacy technicians. We also agree that assessing potential differences between them would be interesting as the focus of a future paper. The goal of the current paper was to improve pharmacy services according to pharmacy professionals’ overall views, regardless of their type or category. We now have added this point in the study’s limitations.

Comment 11 (Reviewer 2): When formatting, put some additional space between the ends of paragraphs before the table title and some additional space below the table. It could help make everything cleaner.

Response 11: We have made the suggested changes.

Comment 12 (Reviewer 2): Like I mentioned above, I don’t know if enough groundwork was laid in the background for all the items being surveyed on within this study. The manuscript could be strengthened by simplifying from all potential added services. Focus on the HDSI services in table titles so it doesn’t seem like other services were just dropped through the survey.

Response 12: All Table titles have been changed and some tables have been removed (also as a response to another Reviewer’s suggestion) for improved clarity.

Comment 13 (Reviewer 2): Simplify results to what is important. 17 tables is hard to keep everything straight and make sure that the question the reader is thinking of is actually the table they are looking to reference.

Response 13: We removed some tables to simplify the reading while keeping the relevant information.

Comment 14 (Reviewer 2): Lines 326 and 327 talk about “several aspects related to service expansion”, but it would be better to put the focus on what based on the background and the service included in the results of the HDSI in focus here.

Response 14: As mentioned in our response to comment 2, the focus of the paper is on new services and service expansion in general, and HDSIs are just an example. Some modifications have been made to the text to make this issue clearer.

Comment 15 (Reviewer 2): There’s a lot of information which was gathered in the survey, to the point that even in the discussion, I don’t believe readers will be able to keep everything straight. All the information either needs to be boiled down to what are the most important factors and simplified that way or needs to be broken up into multiple manuscripts. Is the focus on new services in general and what goes into that or is the focus on this particular new service, because it can go back and forth.

Response 15: We have eliminated parts of the results and have shortened the discussion, following this and another Reviewer’s comments.

Comment 16 (Reviewer 2): Conclusion touches on everything which has been covered in the previous sections of manuscript.

Response 16: Thank you for your comment.

Reviewer 3 Report

A very interesting study looking at a very important topic; please find my comments below as they relate to various sections of the paper

Materials and Methods

1. The participant demographics should be within your result section

2. Line 119 - please indicate what country this was conducted in

2. Regarding your participant list - I am a bit concerned that you combined pharmacy technicians along with pharmacists; I am not sure about the differences in scope between technicians and pharmacists within your country, however, given that their roles may be different and technicians do not have the same clinical education as pharmacists, including them in these results is a bit confusing as they may not participate in these items at the same rate as pharmacists do, so their answers may not be as impactful

3. In Table 1 - please explain what TSDT means

4. When discussing recruiting volunteers, was this done through email?

Results

Overall, the results section is a bit difficult to read; I would encourage you to find a way to perhaps find a way to combine some of the tables and do a quick summary of each instead of listing all of the results that are already found within the table. 

A few grammatical issues throughout the paper but nothing extensive

Author Response

We appreciate the feedback about our manuscript that resulted in valuable improvements. We hope that the responses provided below and changes in the paper (with track changes) can address your concerns.

Kind Regards,

The Corresponding Author

Reviewer 3

Comment 1 (Reviewer 3): The participant demographics should be within your result section

Response 1: Table 1 and participant demographics now appear in the results section, as suggested.

Comment 2 (Reviewer 3):  Line 119 - please indicate what country this was conducted in

Response 2: We have now included the country, as requested.

Comment 3 (Reviewer 3): Regarding your participant list - I am a bit concerned that you combined pharmacy technicians along with pharmacists; I am not sure about the differences in scope between technicians and pharmacists within your country, however, given that their roles may be different and technicians do not have the same clinical education as pharmacists, including them in these results is a bit confusing as they may not participate in these items at the same rate as pharmacists do, so their answers may not be as impactful

Response 3: The goal of the current paper was to improve pharmacy services according to the views of pharmacy professionals in general, regardless of their type or category, as long as they had work experience (namely, interacting with the public). We have added these aspects to the inclusion criteria, for improved clarity, and also discuss this point in the study’s limitations.

Comment 4 (Reviewer 3): In Table 1 - please explain what TSDT means

Response 4: The term TSDT (another designation for pharmacy technician) has been eliminated and is now categorized as “pharmacy technician” for simplification.

Comment 5 (Reviewer 3): When discussing recruiting volunteers, was this done through email?

Response 5: Yes. We have added this information to the text (under Procedure).

Comment 6 (Reviewer 3): Overall, the results section is a bit difficult to read; I would encourage you to find a way to perhaps find a way to combine some of the tables and do a quick summary of each instead of listing all of the results that are already found within the table.

Response 6: We have deleted and combined some tables to address this issue.

4. Response to Comments on the Quality of English Language

Point 1 (Reviewer 3): A few grammatical issues throughout the paper but nothing extensive

Response 1: The text was revised to assure the highest grammatical quality.

5. Additional clarifications

Round 2

Reviewer 2 Report

Very good work on revisions. Thank you for your work on this and allowing us the opportunity to review it.

Reviewer 3 Report

Thank you for your clarifications and updates!